# New Progress in Zebrafish Liver Tumor Models: Techniques and Applications in Hepatocellular Carcinoma Research

**DOI:** 10.3390/ijms26020780

**Published:** 2025-01-17

**Authors:** Qizhuan Lin, Libo Jin, Renyi Peng

**Affiliations:** Institute of Life Sciences, Biomedicine Collaborative Innovation Center of Zhejiang Province, College of Life and Environmental Science, Wenzhou University, Wenzhou 325035, China; 23461044010@stu.wzu.edu.cn (Q.L.); 20160121@wzu.edu.cn (L.J.)

**Keywords:** zebrafish model, liver tumor, cancer research, gene editing

## Abstract

Liver tumors represent a serious clinical health problem that threatens human life. Previous studies have demonstrated that the pathogenesis of liver tumors is complex and influenced by various factors, highlighting limitations in both basic pathological research and clinical treatment. Traditional research methods often begin with the discovery of phenomena and gradually progress to the development of animal models and human trials. Among these, liver tumor animal models play a critical role in advancing related research. The zebrafish liver closely resembles the human liver in structure, function, and regenerative capacity. Additionally, the high transparency and rapid development of zebrafish embryos and larvae make them ideal model organisms for studying liver tumors. This review systematically summarizes recent methods for constructing zebrafish liver tumor models, including transplantation, transgenesis, induction, and gene knockout. Furthermore, the present paper explores the applications of these models in the study of liver cancer pathogenesis, metastasis, the tumor microenvironment, drug screening, and other related areas. By comparing the advantages and limitations of various models and integrating their distinct characteristics, this review provides insights for developing a novel liver tumor model that better aligns with clinical needs. This approach will offer valuable reference information for further in-depth studies of the pathological mechanisms of liver tumors and the development of new therapeutic drugs or strategies.

## 1. Introduction

Hepatocellular carcinoma (HCC) is a malignant tumor that originates in the liver and is the sixth most common malignancy worldwide. Currently, treatment options for liver tumors are inadequate. According to a World Health Organization (WHO) survey in 2020, the mortality rate associated with liver tumors ranks third among the causes of death from malignant tumors. HCCs can be categorized into several pathological types, including HCC, cholangiocarcinoma, and mixed hepatocellular–cholangiocarcinoma, with HCC being the most prevalent, accounting for 75–85% of all primary liver cancers globally. In China, this proportion is even higher, reaching nearly 93%. Risk factors for HCC include chronic infections with hepatitis B and C viruses, alcohol addiction, metabolic liver diseases—particularly nonalcoholic fatty liver disease—and exposure to dietary toxins such as aflatoxins and aristolochic acid [1]. Notably, variations in etiology, tumor growth kinetics, metastatic propensity, and prognosis can arise within the same cohort of patients with HCC due to diverse inducing factors. Therefore, a comprehensive understanding of the underlying causes and mechanisms governing metastasis and proliferation is crucial for tailoring personalized treatment strategies and prognostic assessments in HCC management. In this context, the use of diverse animal models is essential, facilitating profound exploration into the pathogenesis of HCCs.

Animal models are widely utilized in biological research because of their distinct characteristics in morphology, genetics, physiology, and behavior. These animals share numerous similarities with humans and serve as crucial tools in biological and medical studies. Common animal models for HCC include mice, rats, rabbits, macaques, and pigs. Each model has advantages and disadvantages, and no single model can completely replicate human clinical symptoms [2]; thus, various animal disease models should complement each other’s strengths and weaknesses (Table 1).

In recent years, zebrafish have emerged as a novel model offering unique advantages, such as a short breeding time, small space requirements, low cost, and a transparent body structure that facilitates the observation of transplanted cells without an immune response from embryos [11]. This makes zebrafish an effective supplement to traditional animal models. Compared with that of rodents, the transparency of zebrafish embryos and larvae provides unique advantages for real-time high-resolution imaging at both the whole-body and cellular levels in vivo, enhancing the observation of cancer cell proliferation and metastasis. However, similar to mice and humans, adult zebrafish have limited spatial resolution due to the turbidity of their skin and subcutaneous structures. Current in vivo imaging technologies, such as bioluminescence, positron emission tomography (PET), computed tomography (CT), and magnetic resonance imaging (MRI), face challenges in terms of resolution, sensitivity, or the need for expensive equipment [12]. Nonetheless, advancements in biotechnology have led to the development of specific zebrafish models that address these shortcomings. For example, casper zebrafish—a highly transparent adult zebrafish model—and a double transgenic model combining the nacre mutant and roy orbison (roy) zebrafish exhibit nearly complete transparency in adults [12]. Furthermore, the use of isoetharine to inhibit melanin synthesis in zebrafish can also produce transparent specimens, which is particularly useful for imaging the eyes [13].

Although zebrafish and humans have different body structures, their liver functions are highly similar [14]. Comparative analyses of the transcriptomes of hepatocytes from zebrafish, mice, and humans show that the orthologous cell types in zebrafish are completely conserved, with functions closely resembling those of humans [15]. The experimental results indicate that the incidence of HCC induced by oncogenes in zebrafish models approaches 100% [16]. The zebrafish liver tumor model is invaluable for studying the early processes of tumor occurrence and long-term tumor development. As related research has progressed, an increasing number of zebrafish models constructed through various methods have been applied to specific experiments. This review aims to summarize, classify, analyze, and explore studies of zebrafish liver tumor models in recent years, providing a fresh perspective on liver cancer research.

## 2. Application of the Zebrafish Liver Tumor Model in Different HCC Exploration Scenarios

The zebrafish model for studying liver cancer has been successfully utilized to explore and evaluate various aspects of carcinogenesis, including tumor growth, tumor focus formation, proliferation, cancer stem cell dynamics, metastasis (migration, dissemination, extravasation, and infiltration), angiogenesis, and the tumor microenvironment (Figure 1). It is worth noting that the zebrafish model breaks through some of the limitations of traditional models in research. For example, the dynamic observation of cancer cell metastasis in the zebrafish transplantation model, the repeated induction of single transgene and inducer in the same individual, and the dynamic transformation study of liver disease and liver tumors make the study of the zebrafish model more feasible.

### 2.1. Mechanisms of Occurrence and Metastasis of HCC

The occurrence of tumors can be clearly divided into three independent stages: tumor formation, progression, and metastasis. The most common gene mutations associated with tumor formation include *TERT* (telomerase reverse transcriptase) promoter mutations, *TP53*, *CTNNB1* (Catenin Beta 1), *AXIN1*, *ARID1A*, *ARID2*, and other mutations accounting for less than 10% of mutations [17]. Current studies using zebrafish models have successfully demonstrated that genes such as *TP53* [18], *CTNNB1* [19], and *ARID1A* [20] play key roles in the development of HCC. However, the phenotypes associated with the *AXIN1* gene in zebrafish models are not directly linked to liver tumors. For example, mutations in the GSK3 binding domain of zebrafish *Masterblind*/*Axin1* result in reduced or absent eyes and telencephalon, alongside an expansion of telencephalic fates to the anterior part of the brain [21]. Additionally, increased maternal expression of *Axin1* and *Axin2* contributes to the ventralization phenotype observed in the ichabod zebrafish mutant [22]. During HCC progression, several pathways are involved, including telomere maintenance, the P53 pathway, the Wnt pathway, the cell cycle, the Ras/MAPK pathway, and the AKT/mTOR pathway, as well as epigenetic and chromatin remodeling processes, oxidative stress, and angiogenesis [17]. The zebrafish model can effectively reproduce the effects of most of these pathways on liver tumors, highlighting its importance in studying cell signaling pathway activation.

Cancer cell migration within the body is a significant factor contributing to poor clinical prognosis [23]. Given the complexity of metastasis, it is essential to utilize experimental models that can represent and manipulate each step of the metastatic process individually [24]. For example, *TWIST1* is one of the key genes involved in epithelial-to-mesenchymal transition (EMT), which is critical in embryonic development, tumor progression, and metastasis. However, the role of the *TWIST1* gene in liver tumor metastasis in vivo remains unclear [23]. In a study by Jeng-Wei Lu et al., two double-transgenic zebrafish models, *twist1a*+/*kras*+ [23] and *twist1a*+/*xmrk*+ [25], were constructed to investigate how crosstalk between signaling pathways can enhance the migration and dissemination of cancer cells. In recent years, various factors influencing cell migration in vivo have been validated in zebrafish models, including the knockdown of *MAT2B* in cells [26], the downregulation of p-Met and p-AKT protein expression [27], the activation of β-catenin ubiquitination, and proteasomal degradation [28], as well as the overexpression of *LGR5* [29] and induction of *TAp73β* expression [30]. All these factors can significantly impact the migration and spread of cancer cells in vivo. In future research on cancer cell metastasis and invasion, the zebrafish model remains a valuable animal model to consider.

### 2.2. Research on the Mechanism of Spontaneous Regression in Liver Cancer

The regression of liver tumors in clinical practice may be closely related to the treatment methods employed [31]. Common treatment options include surgical resection, chemotherapy, radiotherapy, targeted therapy, and ablation therapy. These approaches aim to eliminate tumor cells and can promote the gradual regression of tumors. In 2002, medical journals published an average of more than four papers per month on spontaneous cancer regression [32]. However, the reported cases are often difficult to reproduce in clinical practice [33,34]. In several liver tumor models induced in transgenic zebrafish, similar phenomena have been consistently observed. When the chemical inducer is withdrawn, the liver tumors in the zebrafish model tend to regress. After several weeks of tumor regression, the tumor-affected livers often revert to histologically normal livers, and tumors can be reinduced by reexposing the fish to the inducing agent, thereby reactivating oncogene expression [35].

The mechanisms underlying tumor regression in different transgenic zebrafish models may vary. (1) Tumor liver cells can be eliminated through cell death, inhibiting tumor cell pathways and allowing normal liver cells to eventually replace tumor liver cells through proliferation and differentiation. In a study by Anh Tuan Nguyen et al., withdrawal of mifepristone resulted in tumor regression in transgenic fish by inducing cell death and inhibiting Ras through targeting its downstream effectors, including the Raf–MEK–ERK and PI3K–AKT–mTOR pathways, which drive liver tumorigenesis [36]. (2) Tumor liver cells may directly revert to normal liver cells. Research by Zhen Li et al. revealed that in addition to the complete replacement of HCC cells by newly generated normal hepatocytes, intermediate phenotypes such as adenomas or hyperplasia were observed in many regression samples, indicating that during tumor regression, some HCC cells can be directly restored to normal liver cells [37]. These findings suggest that some liver tumors may rely on mutations in a single gene for initiation and development. By inhibiting key oncogenes or oncogenic pathways, it may be possible to revert tumor cells back to a normal state.

### 2.3. Liver Tumor Microenvironment

Abnormal lipid accumulation in hepatocytes increases oxidative stress and leads to lipotoxicity, triggering liver inflammation, which is a hallmark of the progression from nonalcoholic fatty liver disease (NAFLD) to HCC. Immune cell populations, including neutrophils and macrophages, provide growth factors, matrix remodeling factors, and inflammatory mediators to the tumor microenvironment (TME), facilitating tumor growth [38]. Current research indicates that both the innate and adaptive immune systems play key roles in the progression of HCC. Further in-depth research is needed to analyze immune responses and liver tumorigenesis in living intact animals. The optical clarity and genetic tractability of zebrafish larvae make them an attractive model for studying the liver microenvironment and immune cell composition using noninvasive fluorescence in vivo imaging [39]. Although the crosstalk between tumors and their local microenvironment has been well studied, the impact of tumors on distant tissues remains understudied [16]. In a study by Yan Li et al., progressive destruction of the intestinal structure was observed in tumor-bearing zebrafish, characterized by villus damage, thinning of the intestinal wall, an increased number of goblet cells, decreased goblet cell size, and eosinophil infiltration, with some fish exhibiting an inflammatory bowel phenotype [16]. This study provides a broader organ-scale view of the liver tumor microenvironment and represents the first systematic characterization of intestinal disruption under liver tumor conditions, targeting intestinal inflammation as a potential approach to managing cancer cachexia.

Research findings regarding the role of neutrophils in tumor progression are conflicting. Some studies highlight the antitumor capabilities of neutrophils in vitro, in vivo, and in clinical contexts, whereas others suggest that neutrophil infiltration may promote tumor progression [40]. For example, Chuan Yan et al. demonstrated that inflammatory signals from oncogenic hepatocytes lead to the rapid recruitment of neutrophils to the oncogenic liver, suggesting that neutrophils promote early liver carcinogenesis [41]. In contrast, Zhen Li et al. reported that neutrophils and macrophages were enriched during both tumor progression and regression. Although these immune cells are relatively evenly distributed in HCC, they accumulate locally during tumor regression, particularly macrophages, which show a dominant vascular association in late regression. These findings suggest that these immune cells may play distinct roles in tumor progression and regression [9]. The mechanistic relationships among different immune cells in the tumor microenvironment are not merely one-way interactions; there may be feedback loops that complicate these dynamics. The zebrafish model offers unique advantages for such research and will be an indispensable tool for uncovering these complex mechanisms.

### 2.4. Drug Screening and Individualized Treatment of Complications

One significant advantage of zebrafish models over rodent models is the ability to conduct high-throughput in vivo chemical screening. This allows researchers to use zebrafish liver tumor models for new drug discovery and preclinical drug screening, evaluating the efficacy and potential toxic side effects of candidate drugs against liver cancer. Over the past 20 years, various studies have demonstrated the therapeutic potential of a range of natural and chemically synthesized products for treating HCC, including aloe base, a 4-phenoxyphenol derivative, MG, TB, bortezomib, and two polytyrosine kinase inhibitors (419S1 and 420S1) [2]. Although there are some limitations in the clinical application of these drugs, this is mainly manifested in physiological differences, drug dose conversion, and drug delivery route differences, so the relevant candidate drugs need to be further verified, such research has the potential to open new avenues for the development of specific therapies. In the study by Marlon E.P. Rosa and colleagues, it was found that anacardic acid (2-hydroxy-6-alkylbenzoic acid, AA) exhibits hydrophobic properties, making it challenging to deliver in aqueous systems. To address this, a natural polymer was developed as an active encapsulation matrix in zebrafish studies. In vivo experiments demonstrated no acute toxicity or changes in locomotor activity in zebrafish treated with this formulation [42]. Through these screening techniques, new angiogenesis inhibitors, such as IROFULVEN and endostatin-mimetic peptide 011, have been identified [9].

Another promising direction for future research is the use of different induced zebrafish models to conduct personalized treatment studies based on individual responses to drugs, which could guide treatment options for clinical liver cancer patients. Cancer anorexia cachexia syndrome (CACS) is a multifactorial syndrome associated with tumors characterized by weight loss, anorexia, and generalized muscle and fat mass atrophy, severely impacting treatment efficacy and quality of life [43]. Sarcopenia, a progressive systemic disease of skeletal muscle, leads to accelerated losses in muscle mass and function and is associated with increased adverse outcomes, including falls, functional decline, frailty, and death [44]. Symptoms such as anorexia make it challenging to implement nutritional strategies in other model animals; however, these strategies can be effectively applied in zebrafish. In studies using the *kras^V12^* transgenic zebrafish model, liver tumors were induced using tetracycline or its analogs, resulting in sustained and severe skeletal muscle atrophy, as evidenced by a gradual decrease in the muscle fiber cross-sectional area (MFCSA). Increased food supplementation was found to accelerate liver carcinogenesis and muscle atrophy, demonstrating a strong correlation between HCC and muscle atrophy [45]. Research by Fei Fei et al. revealed that normalizing insulin-like growth factor 1 (Igf1) expression and disrupting leptin signaling could significantly alleviate anorexia, muscle atrophy, and lipoatrophy in a Ras- and *Myc*-driven HCC zebrafish model [43]. Given that cancer cachexia affects up to 80% of patients with advanced solid tumors [45], addressing this issue is crucial. The use of zebrafish models can enhance the design of future clinical trials and help stage cancer patients based on the degree of cachexia. This approach enables the early initiation of nutritional, metabolic, and pharmacological support before severe weight loss occurs [46].

### 2.5. Interference of Sex Hormones on Liver Tumors

Sex differences in primary liver cancer have been well documented, with the incidence in men being 2–8 times greater than that in women [47]. Traditionally, this sex bias has been attributed to men’s increased susceptibility to known risk factors for HCC, such as heavy alcohol consumption and unhealthy diets. However, experiments in laboratory mice have also demonstrated that males are more susceptible to HCC than females, with 100% of males and only 30% of females developing HCC under chronic carcinogen exposure [48]. This phenomenon has been replicated in transgenic zebrafish models, prompting further investigation into the mechanisms underlying sex differences associated with HCC.

Numerous studies using zebrafish models have consistently reported these differences. Research conducted by Xiaojing Huo et al. revealed that larval oncogenic hepatocytes exhibit cancer stem cell characteristics. Female oncogenic hepatocytes resemble mild human HCC subtypes, whereas male oncogenic hepatocytes align more closely with severe HCC subtypes, mirroring the sex differences observed in both zebrafish and humans [49]. Further research by Yan Li and colleagues demonstrated that estrogen treatment has tumor-suppressive effects in the early stages of tumor development by inhibiting cell proliferation, whereas androgens promote tumor growth by increasing cell proliferation [50]. The roles of sex hormones in liver tumorigenesis and regression were explored using the inducible double-transgenic zebrafish *Myc*/*xmrk*. Hankun Li et al. treated *Myc*/*xmrk* fish with androgens or estrogens and reported that the male hormone 11-ketotestosterone (KT11) generally stimulated HCC progression in female *Myc*/*xmrk* fish and delayed tumor regression. Conversely, estrogen (E2) delayed HCC progression and accelerated regression in male *Myc*/*xmrk* fish, indicating that sex hormones significantly influence both tumor progression and regression, contributing to the observed sex differences [51]. Although current research confirms that sex hormones affect liver tumor diseases, it also highlights that the tumorigenic impact of these hormones is limited, as animal models continue to develop HCC despite hormonal interventions. More relevant research is needed to enhance the clinical applicability of these findings.

### 2.6. Research on the Impact of Genetic and Environmental Factors on Liver Cancer

Genetic and environmental factors play crucial roles in the occurrence of liver cancer. The significance of hMOF (human males absent on the first, KAT8) in vascular invasion of HCC is highlighted in the study by Nicolas Poté et al., which supports the transcriptional activation of key genes involved in this process [52]. These findings underscore the major role of epigenetic changes in HCC progression and provide valuable insights for future targeted therapies. The zebrafish model is particularly useful for exploring the impact of environmental factors, including chemicals, nutritional influences, and toxins, on liver cancer. While mammalian models often provide insights into the stress effects of environmental factors on humans, zebrafish offer unique advantages in certain studies. Given that a significant proportion of tumors are thought to originate from environmental exposures, the potential carcinogenic risks associated with these factors must be thoroughly assessed, as outlined by the International Agency for Research on Cancer (IARC) [53]. The complexity of potential environmental factors necessitates experimental designs that can address the interactions between multiple elements. Owing to their efficiency and rapid development, zebrafish models are particularly well suited for this purpose. In fact, zebrafish have emerged as one of the primary model organisms for the preliminary screening of environmental carcinogens, making them valuable tools in cancer research. The earliest environmental factors studied in zebrafish mainly focused on pollutants of global concern, such as persistent organic pollutants (POPs) [54] and heavy metals [55], and then gradually expanded to endocrine disrupters [56], new pollutants [57], and physical factors.

## 3. Construction of a Zebrafish Liver Tumor Model Through Transplantation Methods

### 3.1. Transplantation Methods

Currently, the predominant method for constructing liver cancer model animals is xenotransplantation, which is widely used and has a high success rate [58]. Xenograft models of liver cancer closely resemble human cancer cell lines at the cellular level. By implanting tumor cells or tissues marked with fluorescent proteins into model animals, researchers can observe the proliferation and metastasis of cancer cells in vivo, aiding in predictions of human liver cancer development, treatment efficacy, and drug screening (Figure 2). Notably, zebrafish have low-level or absent immune systems during their embryonic and early juvenile stages, making them particularly amenable to xenograft models and more receptive to transplanted cells. This characteristic has garnered significant attention for xenograft zebrafish models, leading to rapid advancements in related research fields [59]. However, these models have limitations, including potential trauma responses from implantation surgery and anatomical differences between zebrafish and mammals, which need to be addressed for more accurate results.

### 3.2. Embryo Transfer Model

Zebrafish serve as a high-throughput in vivo model, enhancing success rates in later stages of preclinical drug development and reducing the economic and time costs associated with the screening process. Among the animal models constructed through transplantation methods, zebrafish models offer several significant advantages in the larval stage compared with rodents. Additionally, the unique immune deficiency stages of zebrafish eliminate the need for immunosuppressive treatments when xenograft models are constructed, thereby reducing potential interference in the construction process. In a study by Federica Tonon et al., a novel, rapid, and cost-effective zebrafish HCC xenograft model was developed by microinjecting the HCC cell line JHH6 into the yolk sac of zebrafish embryos for drug screening purposes [27]. An increasing number of researchers have recently explored the use of microinjection to transfer liver cancer cells into zebrafish embryos, resulting in improved construction standards for these models. Zebrafish models are gaining prominence in the fields of tumor xenograft models, developmental biology, genetics, and cancer research, occupying an increasingly important position in scientific inquiry.

### 3.3. Adult Fish Transplant Model

The embryo transplant model is limited to the early stages of embryonic development, and models constructed at this stage may not guarantee continued growth and development. Consequently, some research may necessitate the use of more mature organs or adult samples. To address this, researchers have explored injecting cells into the tissues or organs of adult zebrafish to create adult fish transplant models of liver tumors. In a study by Jianhong Yang et al., fluorescently labeled HepG2 cells were successfully introduced into the small intersegmental blood vessels of transgenic zebrafish. The results revealed that some tumor cells remained quiescent, whereas others underwent apoptosis or fragmentation. Notably, tumor cells with strong metastatic potential were able to penetrate and extravasate from host blood vessels into adjacent tissues within 24 h [60]. Although the stability of adult zebrafish models constructed using transplantation methods has yet to be fully established, these models represent promising avenues for development and application and could play irreplaceable roles in the study of certain cases of nonspontaneous liver tumors.

### 3.4. Application Effects of Zebrafish Models Constructed Through Transplantation

Dozens of xenograft zebrafish liver cancer models have been developed to date. Models constructed from different cell lines and injection sites exhibit distinct characteristics and varying applicability. Recent experimental studies have shown that yolk sac microinjection is the most successful method for constructing these transplantation models. The following sections evaluate the applications and effects of xenograft zebrafish models in related fields over the past five years (Table 2).

## 4. Transgenic Methods for Constructing a Zebrafish Liver Tumor Model

### 4.1. Transgenic Methods

The production of genetically modified organisms is invaluable for studying the genetic basis of embryonic development in both mammalian and invertebrate species [66]. Comparisons with the human reference genome revealed that approximately 70% of human genes have at least one clear zebrafish ortholog, and approximately 85% of disease-causing genes have zebrafish counterparts [67,68]. Due to these similarities, zebrafish are considered promising transgenic research models for studying various human diseases, including genetic disorders. However, the research process has faced challenges. In 1988, the first transgenic zebrafish were successfully created by microinjecting plasmid DNA carrying foreign genes into fertilized eggs [14]. Unfortunately, the transgenic zebrafish produced by this method lacked stability. It was not until 2011 that stable transgenic zebrafish models capable of expressing liver cancer were reported [69]. Currently, researchers have successfully constructed several liver tumor models by expressing transgenes in zebrafish livers, primarily using three common types of oncogenes: *kras*, *xmrk*, and *Myc*. These transgenic zebrafish typically develop hepatocellular adenomas (HCAs) that can progress to HCC and other liver tumors of varying severity. Together, these three zebrafish liver cancer models account for nearly half (47.2%) of human HCC cases, with a significant correlation observed between some human HCCs and these oncogene-addicted zebrafish tumors [70]. Transgenic zebrafish have a high success rate and high stability in the construction of liver tumor models (Figure 3). Owing to their rapid reproduction rate, zebrafish play a crucial role in the early exploration of clinical treatments. However, due to limitations in transgenic technology—such as gene superposition effects—liver tumors induced by specific genes cannot fully represent the relevant pathogenesis. Therefore, additional models are needed to support related research.

### 4.2. kras^V12^ Gene Overexpression Model

Cross-species comparisons of cancer transcriptomes have further elucidated HCC-specific gene signatures and liver cancer progression signatures that are evolutionarily conserved between humans and zebrafish [69]. One of the most significant transgenic zebrafish liver tumor models is the *kras^V12^* gene overexpression model, which involves the transformation of zebrafish liver cells through *kras^V12^* overexpression. This model represents one of the earliest stable transgenic zebrafish liver tumor models. In a study by Anh Tuan Nguyen et al., the molecular mechanisms underlying progressive Ras-induced HCC were explored using the liver-specific fatty acid-binding protein 10 (fabp10) promoter, which drives oncogenic *kras^V12^* expression specifically in zebrafish livers [69]. Ras has been identified as a potent oncogene and a central regulator of multiple signal transduction pathways in human cancer, with activation occurring in more than half of HCC cases [36]. Consequently, the *kras^V12^* overexpression model effectively mirrors many clinical features of liver tumors. This zebrafish liver cancer model excels in various applications, including studies of tumorigenesis, oncogene addiction, tumor microenvironment interactions, sex differences, cancer cachexia, and drug screening. However, although it reveals several molecular mechanisms of Ras-driven liver tumorigenesis and recapitulates typical features of human HCC, the constitutive high-level expression of Ras often leads to early tumorigenesis and premature mortality. In contrast, low-*Krasv12*-expressing strains exhibit delayed tumor onset and a lower incidence of tumors (approximately 30%), which may limit the model’s utility for large-scale research and drug screening [36]. Despite these challenges, ongoing efforts are aiming to develop improved zebrafish liver tumor models, building on the foundational *kras^V12^* model.

### 4.3. xmrk Model

Epidermal growth factor receptor (EGFR) and its ligand EGF have emerged as promising targets for human liver cancer therapy. The *xmrk* gene is a naturally occurring mutation of the EGFR subtype EGFRb found in fish of the genus Xiphophorus (including flatfish and swordtail fish) [43]. This mutation features two alterations in its extracellular domain, leading to constitutive autophosphorylation and activation of downstream signaling pathways [43]. The *xmrk* model employs the Tet-on system to achieve liver-specific expression of the superactive fish oncogene *xmrk*, resulting in the construction of transgenic zebrafish capable of inducing liver cancer [43]. In both juvenile and adult *xmrk* transgenic fish, liver tumors are rapidly induced with 100% penetrance through continuous induction over varying time periods [43]. Remarkably, this model exhibits most of the histological characteristics of human HCC and displays a unique phenomenon of rapid tumor regression once the inducer is removed. This model provides an ideal experimental framework for studying the initiation and maintenance of HCC driven by a single oncogene, paving the way for the development of treatments targeting oncogene addiction or related oncogenic pathways.

### 4.4. Myc Overexpression Model

Both *xmrk* and *Myc* serve as oncogene-related biomarkers for HCC. The *Myc* model utilizes the Tet-on system for liver-specific expression of mouse *Myc*, creating a transgenic zebrafish liver tumor model [71]. This model involves dose-dependent *Myc* expression, which leads to liver hyperplasia that ultimately progresses to hepatocellular adenoma and liver cancer with prolonged induction [71]. Research by Pal Kaposi-Novak et al. indicated that the activation of the *Myc* transcriptional signature is closely associated with the malignant transformation of precancerous liver lesions [72]. The transcriptome of *Myc*-induced zebrafish liver tumors is highly similar to the transcriptomes of various stages of human liver disease, including low-grade dysplastic nodules (LGDNs), high-grade dysplastic nodules (HGDNs), and various stages of HCC (very early HCC, early HCC, late HCC, and advanced HCC) [71]. Like in the *xmrk* model, the transient inactivation of *Myc* due to the cessation of inducers results in tumor regression and the differentiation of cancer cells into mature bone cells [71]. However, reactivation of *Myc* does not restore malignancy but instead induces apoptosis [73]. These findings suggest that the *Myc* overexpression model possesses unique characteristics that differentiate it from other zebrafish liver tumor models, offering potential avenues for developing targeted therapies for specific cancers.

### 4.5. Multitransgenic Model

To date, dozens of inducible transgenic zebrafish have been successfully constructed using native inducible gene promoters or chimeric promoters that incorporate well-characterized responsive cis elements [72]. With advancements in zebrafish transgenic technology, researchers have started to explore the construction of multitransgenic models. Double transgenic technology allows the introduction of two different genes into the target organism, enabling the simultaneous expression of multiple traits and facilitating more complex functional enhancements. For example, the transparency of casper double mutant zebrafish has improved due to the interaction of two genes, allowing for direct visualization of organs such as the heart, intestines, liver, and gallbladder using a standard stereomicroscope [12]. Currently, multitransgenic zebrafish combining fluorescent genes with liver disease genes have also been reported. Research indicates that when there is potential complementarity or synergy between two genes, this effect can be more effectively realized through double transgene technology. For example, a study by Jerry D. Monroe et al. demonstrated that the induction of *xmrk*, *Myc*, and *xmrk*/*Myc* can lead to different stages of HCC. During tumor progression, lipid deposition and grade generally increase, whereas triglyceride levels decrease. In the double transgenic model, *Myc* appears to regulate lipid species levels in early HCC stages, whereas *xmrk* may take over this regulation in later stages, with the complementary effects of the two genes enhancing the speed and extent of tumorigenesis [73]. Additionally, a study by Jeng-Wei Lu et al. revealed that the tumor metastasis rate in *twist1a*+/*kras*+ double transgenic zebrafish was more than 20% greater than that in single transgenic zebrafish, demonstrating that different genes can cooperate to induce tumor metastasis [23].

### 4.6. Application Effects of Zebrafish Models Constructed Through Transgenic Methods

In recent years, as clinical research has advanced, the limitations of single transgenic models have become increasingly apparent, leading researchers to explore the use of multiple transgene combinations to construct more relevant models for research applications. These multitransgenic zebrafish models offer greater complexity and can more accurately replicate human disease processes, allowing for more comprehensive studies of disease mechanisms and therapeutic testing. Below is an overview of recent applications and evaluations of multitransgenic zebrafish models in various research fields over the past five years (Table 3).

## 5. Construction of Zebrafish Liver Tumor Models Through Induction Methods

### 5.1. Induction Methods

Inducible zebrafish liver cancer models have become valuable for verifying cancer-related pathogenesis at the histological, transcriptomic [78], and molecular levels. This approach focuses on both eliminating and preventing cancer factors and constructs models through the use of chemicals, heavy metals, and radiation to simulate carcinogenic environments. Through repeated exposure and subsequent expression, an in vivo model closely resembling human liver cancer characteristics was generated. Early work by JAN M. Spitsberg [79] in the late 20th century using N-methyl-N’-nitro-N-nitrosoguanidine (MNNG) to induce liver tumors laid the groundwork for modern inducible zebrafish cancer models and highlighted their potential in cancer research. Induced zebrafish models not only exhibit clinical parallels to human liver tumors but also serve as early detection tools for environmental carcinogens. They provide insight into the complexities of the impact of carcinogens, simulating the multifactorial progression of liver cancer in a way that reflects real-world exposures. Although conventional induced models have several limitations—namely, difficulty in isolating single cancer pathways owing to the broad effects of carcinogens—they continue to be indispensable. By integrating recent technological advancements, researchers have enhanced the scope and application of these models, enabling their use in diverse aspects of clinical research and reinforcing their role in understanding and combating liver cancer (Figure 4).

### 5.2. Chemical Induction Model

Liver cancer is caused primarily by alcoholism, chronic hepatitis B and C infections, and exposure to environmental toxins [80]. Zebrafish models are induced with carcinogens to mimic liver disease progression into tumors. However, due to the nonspecific nature of carcinogen effects and the physiological and immune differences between zebrafish and humans, these induced models often fail to consistently replicate isolated liver conditions, which complicates their application in liver cancer research. Diethylnitrosamine (DEN), a hepatotoxic and hepatocarcinogenic compound, has been instrumental in the development of HCC models in various species [80,81]. In studies by Jegannathan Srimathi Devi and colleagues, DEN successfully induced liver fibrosis in zebrafish, showing its potential for inducing liver disease in non-mammalian animals [82]. This zebrafish model exhibited key liver pathologies, including fatty tissue lesions, peritoneal effusion, and tumor development in the liver, making it a practical and efficient model for investigating liver cancer.

However, the prolonged induction period and inconsistent lesion progression limit the effectiveness of purely carcinogen-induced models. Recent advancements in transgenic technology have led to combined models in which chemical induction and genetic modifications stabilize liver tumor models in zebrafish. For example, Reboredo et al. explored transgene expression changes in the liver by combining the rtTA2(S)-M2 and GLp65 transactivators with doxycycline and mifepristone for a more precise induction control [83]. Mifepristone, an antiprogestogen agent, was also used to regulate gene expression via the *LexPR* system, increasing the model’s specificity [84]. These combined inducible zebrafish models have become pivotal in liver cancer research, offering stable lesion progression and accurate control of life cycle expression. Their applications have expanded to include drug screening, and some models have demonstrated the potential for tumor regression—a rare phenomenon in clinical settings. Understanding the mechanisms behind this regression could unlock new pathways for cancer treatment, making these models valuable for future explorations in liver cancer therapy.

### 5.3. Dietary Induction Model

The pathogenesis of HCC involves a progressive sequence of liver damage starting with hepatitis or fatty liver disease, advancing to fibrosis, and finally to cirrhosis, which can evolve into HCC over decades [9]. Researchers have leveraged zebrafish to model this stepwise progression by establishing both alcoholic liver disease (ALD) and diet-induced obesity (DIO) models, simulating aspects of NAFLD. These models provide valuable insights into the mechanisms of primary liver tumor development.

Zebrafish larvae are especially suitable for studying ALD due to their ability to metabolize alcohol, with the liver reaching maturity at four days postfertilization (DPF). ALD can be induced by adding ethanol to the larval breeding environment, effectively inducing alcohol-related liver damage. However, a key challenge in this model is maintaining sufficient food intake in ethanol-exposed larvae, a difficulty that prevents the sustained induction of ALD, combined with the progressive development of HCC—a challenge that remains for future research. NAFLD, which encompasses simple steatosis to NASH with chronic inflammation, can lead to fibrosis, cirrhosis, and eventually HCC [24]. In zebrafish, diet-induced obesity models provide a robust approach to studying NAFLD by simulating human dietary habits. Feeding zebrafish a high-sugar or high-fat diet induces NAFLD-related diseases, and with extended exposure, disease severity increases, including the risk of related comorbidities. These diet-induced zebrafish models closely replicate the primary pathogenesis of hepatocellular tumors in humans, offering an accessible and representative platform for investigating dietary impacts on liver disease progression.

### 5.4. Application Effects of Zebrafish Models Constructed Through Induction Methods

Induced zebrafish liver cancer models, developed through exposure to carcinogens, provide an essential platform for clinical research because of their alignment with the pathogenesis of human liver diseases. These models offer unique insights into disease progression by replicating the stages and mechanisms of HCC development observed in human cases, from initial hepatic damage to tumor formation. Despite the variability in their construction compared with those of genetically stable models, such as transgenic approaches, their relevance to environmentally and chemically induced liver cancer makes them a cornerstone in the study of clinically relevant disease pathways. Recent advancements in zebrafish modeling have led to the development of standardized protocols that improve the consistency and reproducibility of induced models. These standards focus on optimal carcinogen types, dosages, and exposure durations, as well as postexposure care, which have collectively enhanced the reliability and applicability of induced liver cancer models in research. This rigorous approach enables researchers to simulate pathophysiological changes leading to liver cancer more accurately, creating a bridge between experimental models and human clinical conditions. In the last five years, induced zebrafish models have become increasingly instrumental in assessing environmental carcinogens, examining potential chemopreventive agents, and understanding the molecular underpinnings of liver carcinogenesis. The following sections detail the application and impact of these models in recent research, highlighting their contributions to advancing our understanding of liver cancer etiology and intervention strategies (Table 4).

## 6. Gene Knockout Methods for Constructing Zebrafish Liver Tumor Models

### 6.1. Gene Knockout Methods

Gene knockout technology enables targeted manipulation of endogenous DNA, reducing or eliminating the expression of specific proteins [86]. This technique is invaluable for constructing precise animal models that reflect the mechanisms of human diseases, especially in the study of genetic and complex polygenic diseases. To achieve a reliable model, factors such as accuracy, reproducibility, cost-effectiveness, and usability across research fields are crucial [9]. Recent advancements in gene-editing tools, including zinc finger nucleases (ZFNs) [31], transcription activator-like effector nucleases (TALENs) [32], and CRISPR/Cas systems [34], have transformed model development, with CRISPR/Cas9 being especially effective and widely adopted in zebrafish research [33]. CRISPR/Cas9 has facilitated rapid and precise gene knockout, allowing researchers to silence specific tumor suppressor genes, such as *Tp53* and *PTEN*, which are implicated in the pathogenesis of HCC. Through this technology, researchers can investigate how gene deletions contribute to liver cancer development in zebrafish, providing insights into gene function and disease mechanisms (Figure 5). Gene knockout zebrafish models offer high potential for understanding the genetic factors involved in liver cancer and other diseases; however, challenges remain. Off-target effects and the unintended silencing of genes across generations can limit precision. Additionally, compensatory mechanisms within organisms may offset the effects of gene knockouts, potentially confounding results. Due to these limitations, knockout zebrafish models are still emerging as research tools and are used selectively in studies requiring intricate analysis of gene–disease interactions.

### 6.2. p53 Mutation Model

The *p53* tumor suppressor gene is essential for regulating apoptosis, cell cycle arrest, and DNA damage repair, with mutations leading to its inactivation in approximately 50% of human cancers [47]. In the liver, *p53* ablation exacerbates liver pathology and polyploidization, even though it does not significantly affect other tissues, such as the kidney, germ cells, or brain [48]. Initial studies with *p53*-mutant zebrafish models did not reveal tumor formation in liver tissue from *p53* mutation alone. However, research by Jeng-Wei Lu et al. demonstrated that combining HBx (hepatitis B virus X protein) with *p53* mutations induces HCC in zebrafish through the activation of the src pathway, with HBx inducing tumorigenesis only in the presence of *p53* mutations [49]. This HBx/*p53* mutant model mirrors aspects of human HCC, providing a valuable in vivo platform for potential liver cancer drug screening. The zebrafish *p53*-HBx model represents a major step toward replicating human HCC, but the widespread roles of *p53* in stem cell proliferation, differentiation, and overall development complicate its use. No current model can entirely eliminate the growth variations introduced by *p53* mutations, which limits the model’s reliability and application scope in research [44]. Therefore, although promising, these models require further refinement for broader applications.

### 6.3. PTEN Knockout Model

The *PTEN* tumor suppressor gene, known for its dual protein and lipid phosphatase activities, is one of the most commonly mutated genes in sporadic cancers [45]. Its tumor-suppressive properties rely mainly on its lipid phosphatase function, which negatively regulates the PI3K–AKT–mTOR signaling pathway [45]. In a study by Juanjuan Luo et al., researchers used the CRISPR/Cas9 system to create a transgenic zebrafish strain with *PTEN*-specific mutations to genetically induce liver tumorigenesis [46]. The results showed that *PTEN* loss alone was insufficient to induce liver cancer in zebrafish, highlighting that *PTEN* depletion alone may not always trigger tumorigenesis in certain tissues. Further investigations by Luo’s team employed CRISPR/Cas9 to generate zebrafish models with dual mutations in *PTEN* and *p53*. These models presented greater incidences of tumors, higher histological grades, and shorter survival times than single-gene mutation models did, suggesting a synergistic interaction between the *PTEN* and *p53* pathways in promoting tumor initiation and progression [46]. This dual mutation model underscores the cooperative role of *PTEN* and *p53* in hepatocarcinogenesis in zebrafish, providing a compelling basis for a more detailed exploration of the mechanistic pathways underlying liver cancer and potential therapeutic targets.

### 6.4. Application Effects of Zebrafish Models Constructed Through Gene Knockout Methods

In the early stages of zebrafish liver cancer model development, the limitations of gene knockout technology pose challenges, restricting the application of knockout models. However, recent advancements in gene editing methods, including CRISPR/Cas9, TALENs, and ZFNs, have revitalized interest in these models, enabling a more precise and effective knockout of specific genes. Current studies suggest that silencing or reducing the expression levels of certain tumor suppressor genes, such as *p53* and *PTEN*, can play a critical role in liver tumor development. This aligns with findings from knockout models, which help to delineate how individual gene loss contributes to cancer initiation and progression. Compared with other construction methods, knockout zebrafish models tend to focus on the specific loss of one or a few genes, somewhat limiting their scope. However, they are invaluable in pinpointing the specific roles of genes in cancer pathology. These models are particularly useful for investigating the functional impact of gene deletions on tumor development, making them ideal for mechanistic studies in liver cancer research. The following is an application and effectiveness evaluation of induced knockout zebrafish models in liver cancer research from recent years (Table 5).

## 7. Conclusions

Research on zebrafish liver tumor models provides a powerful tool for exploring the pathogenesis of liver cancer and preclinical drug screening. Zebrafish liver cancer models constructed through transplantation, transgenesis, induction, gene knockout, and other methods have shown great potential in terms of tumor formation, development, metastasis, and treatment response, such as the following: (1) xenotransplantation has a low immune expression stage and can be used to conduct in-depth research on the invasiveness of liver cancer cells; (2) models combining transgenes and drug induction help to explore the unique spontaneous regression mechanism of zebrafish liver tumors; and (3) zebrafish models facilitate real-time observation of tumor growth and metastasis processes and are suitable for high-throughput screening of potential therapeutic drugs. However, these models also have limitations such as certain biological differences, differences in metabolic pathways, and poor prediction ability for some drug responses. Therefore, future research should consider combining the advantages of multiple animal models to compensate for the shortcomings of a single model and further improve the clinical translation potential of liver cancer research. For example, the zebrafish model for candidate drug screening combined with a mouse model and cell in vitro validation can greatly shorten the development cycle of new drugs. Future research should also strengthen the application of zebrafish models in personalized treatment, especially in high-throughput drug screening, tumor microenvironment research, and the treatment of complications. By continuously optimizing the construction method of zebrafish models and promoting the development of gene editing technology, more precise treatment strategies and better prognosis management for patients with liver cancer are expected.

## Figures and Tables

**Figure 1 ijms-26-00780-f001:**
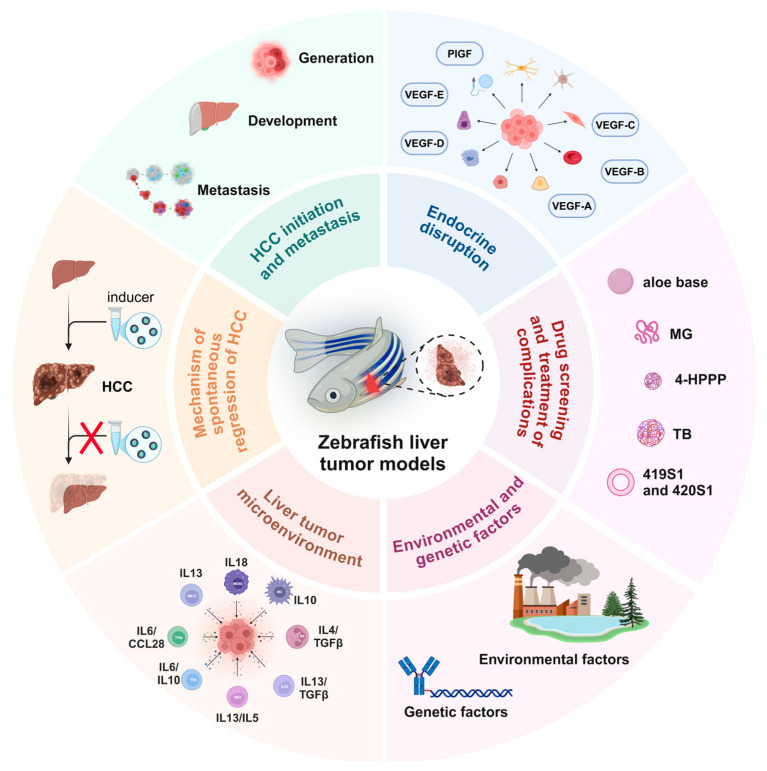
Schematic diagram of the application of the zebrafish liver tumor model in different liver cancer exploration scenarios. 4-HPPP: 4-phenoxyphenol derivative; MG: methyl gallate; TB: teabrownin.

**Figure 2 ijms-26-00780-f002:**
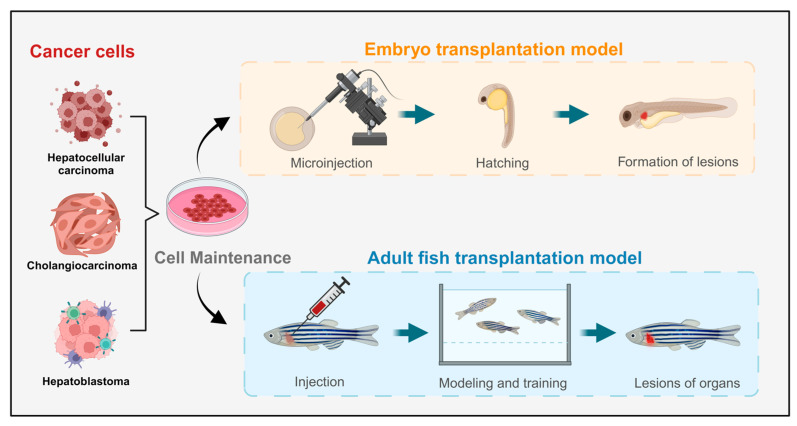
Schematic diagram of the liver tumor zebrafish model constructed through transplantation methods.

**Figure 3 ijms-26-00780-f003:**
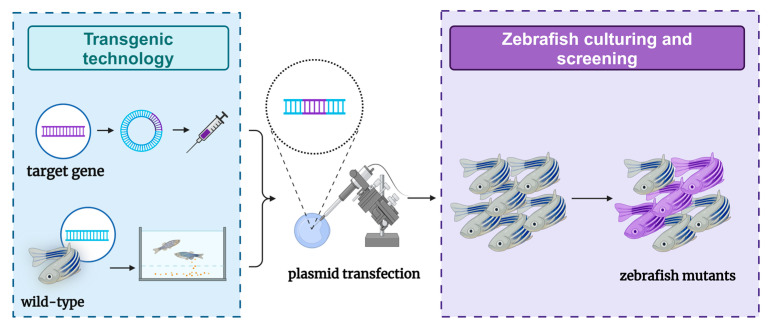
Schematic diagram of the zebrafish model of liver tumors constructed through transgenic methods.

**Figure 4 ijms-26-00780-f004:**
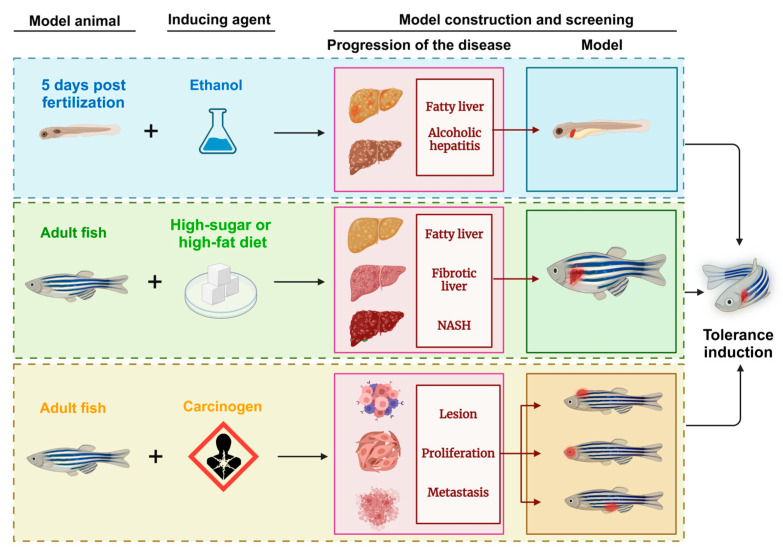
Schematic diagram of the zebrafish model of liver tumors generated through induction methods. NASH: nonalcoholic steatohepatitis.

**Figure 5 ijms-26-00780-f005:**
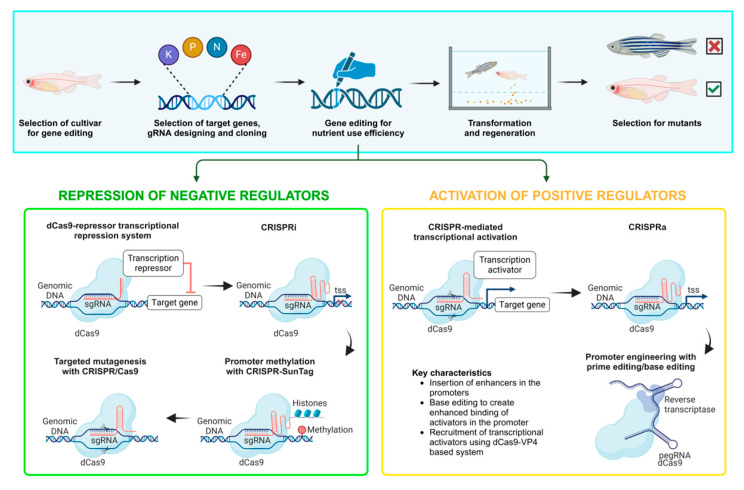
Schematic diagram of zebrafish models of liver tumors generated through knockout methods. sg RNA: single-guide RNA; TSS: transcription start site.

**Table 1 ijms-26-00780-t001:** Comparison of the advantages and disadvantages of different liver cancer animal models.

Category	Model Animal	Advantages	Disadvantages	Main Applications	References
Mammals	Mouse	Similar genetics to humans, mature gene editing	Liver structure and cancer microenvironment differ from those of humans	Liver cancer mechanisms, tumor growth, new drug screening	[3]
	Rat	Closer to human structure, suitable for complex surgeries and toxicology studies	Challenging gene editing, high cost	Liver cancer mechanisms, tumor treatment	[3,4]
	Rabbit	Moderate size, structure similar to humans	Limited gene editing, low liver regeneration	Drug metabolism, toxicity studies	[5]
	Rhesus monkey	Genetic and biological characteristics close to humans	Ethical and cost limitations	Liver cancer mechanisms, drug evaluation	[6]
	Pig	Liver size and function similar to humans	High cost, slow reproduction	Liver physiology, surgery, transplantation	[7]
Non-mammals	Zebrafish	Small size, fast reproduction, transparent embryos, rapid gene editing	Different liver structure, does not fully replicate human liver processes	Liver cancer mechanisms, drug screening	[8,9,10]

**Table 2 ijms-26-00780-t002:** Application and evaluation of the zebrafish model constructed through transplantation methods.

Cell Line	Injection Method	Application	Main Outcome Assessment	References
Hep 3B2.1-7 and Li-7	Microinjection	ADI efficacy against HCC	ADI inhibits BIRC5, FEN1, and the EGFR/PI3K/AKT signaling pathway	[61]
HepG2	Microinjection	HCC cell migration rate	*MAT2B* knockdown reduces HCC cell migration	[26]
HepG2	Microinjection (yolk sac)	Sorbaria sorbifolia (SS) anti-tumor efficacy	SS downregulates p-Met and p-AKT, inhibiting tumor growth	[27]
SK-Hep-1	Microinjection (yolk sac)	Effect of TB on HCC	TB induces apoptosis in cancer cells	[9]
Huh7 (EGFP overexpressed)	Microinjection (yolk sac)	Src and PARP1 combination therapy	Src and PARP1 inhibitors show synergistic lethal effects on HCC	[60]
Hep3B_Lifeact-RFP	Microinjection (yolk sac)	Impact of With-no-lysine (K)-1 (WNK1) overexpression	WNK1 knockdown reduces tumor angiogenesis and cell proliferation	[62]
BEL-7402	Microinjection (yolk sac)	NF-κB and c-JUN synergy verification	NF-κB and c-JUN are potential HCC therapy inducers	[63]
HepG2-HBx	Microinjection	Anti-metastatic effect of the compound Phyllanthus urinaria L. (CP)	CP inhibits metastasis in HBV-related HCC	[64]
HuH-7-Lgr5	Microinjection (yolk sac)	Metastatic potential of LGR5 cells	LGR5 overexpression enhances metastatic potential	[65]
Hep3B-TAp73β	Microinjection (yolk sac)	TAp73β impact on HCC cell migration	TAp73β induces a twofold increase in migration ability	[37]

**Table 3 ijms-26-00780-t003:** Application and evaluation of the transgenic zebrafish model.

Gene	Construction Method	Specific Findings	Outcome Assessment	References
*CTNNB1*	Phosphorylation site mutation (4-point)	Activates β-catenin, influences specific lipid metabolism	Common oncogenic mutation in 30% of HCC tumors	[19,74]
*CD36*	Inject expression vector with Tol2 transposase mRNA	Anti-HCC effect of oligofucose polysaccharides in *CD36* model	Anti-HCC, anti-steatosis, anti-fibrosis	[75]
*CD36*/*tert*	Microinjection and hybridization	Anti-HCC effect of Carassius auratus complex formula (CACF) dose	Anti-HCC in zebrafish xenograft model	[76]
*kras^V12^*	Mutant allele introduction into EGFP-*kras^V12^* zebrafish	Affects HCC stress response and liver growth	Mutation reduces liver cancer cell growth and survival	[77]
*Xmrk*	Tet-on system for transgenic zebrafish	Rapid HCC regression upon inducer withdrawal	*Xmrk* model shows tumor spontaneous regression	[43]
*twist1a*+/*kras*+	Doxycycline and 4-hydroxytamoxifen induction	Significant role of LPS in double-transgenic zebrafish	Lipopolysaccharide (LPS) may exacerbate HCC metastasis	[23]
*twist1a*+/*xmrk*+	Same as above to create double-transgenic zebrafish	High-dose Dox induces liver tumor metastasis	Stronger metastatic ability in transgenic zebrafish	[25]
*Myc* and *Ras*	Hybrid generation of double-transgenic zebrafish	Exhibits anorexia/cachexia-like phenotype	Severe muscle atrophy in Tg (*Myc* and *Ras*)	[43]
*Tg (Myc* and *Ras Mosaic)*	Random UAS promoter silencing for new transgenic model	Transforms the entire liver, simulating real HCC heterogeneity	Stable muscle and fat atrophy in Tg (*Myc* and *Ras Mosaic*)	[43]
*CreER*/*xmrk*	Cre/loxP method for new transgenic strain	Single oncogene induction triggers tumors	Dox and 4-Hydroxytamoxifen(4-OHT) treatments achieved time control of Cre-mediated recombination	[35]
*Myc*/*xmrk*	Hybrid double-transgenic zebrafish model	Androgen KT11 stimulates HCC progression	Significant sex-based differences in HCC progression	[51]

**Table 4 ijms-26-00780-t004:** Application and evaluation of induction methods for constructing zebrafish models.

Inducer	Construction Method	Application	Outcome Assessment	References
Ethanol	Ethanol-induced zebrafish	Alcoholic liver disease (ALD) model	Acute effects reversible, larvae show feeding difficulties, no advanced fibrosis	[14]
Mifepristone	*LexPR* system for liver-specific EGFP-*kras^V12^* expression	Tumor progression and regression in liver	Applicable for tumor progression analysis and anticancer drug screening	[36]
Fructose	4% fructose in culture medium	Induces NAFLD, progresses to HCC	Poor repeatability and stability for primary liver tumor induction in NAFLD model	[24]
Doxycycline	Tetracycline-inducible *kras^V12^* transgenic zebrafish	Study of sex differences in HCC development	Examines perfluorooctane sulfonate effects on male HCC progression	[85]
High-Fat Diet (HFD)	High-cholesterol diet from days 5–12	Angiogenesis in HCC	Short-term HFD induces malignant cell and nuclear changes	[40]

**Table 5 ijms-26-00780-t005:** Application and evaluation of gene knockout methods for constructing zebrafish models.

Gene	Construction Method	Application Direction	Outcome Assessment	References
*Pten*/*tp53*	Gene knockout using CRISPR/Cas9 system	Study the roles of the *Pten* and *Tp53* pathways in liver cancer	High tumor incidence and severe clinical symptoms in double-mutant zebrafish	[46]
*Tg(fabp10a*, *src*, *p53)*	Triple transgenic zebrafish with diet-induced obesity	Evaluate the impact of 4-Aminobiphenyl (4-ABP) on liver cancer	Activation of the Ras–ERK pathway observed with repeated 4-ABP exposure, promoting liver cancer	[18]
*tp53*−/−	Hybridization of *Myc*AG fish and *tp53*−/− mutants, PCR genotyping	Study the role of *Myc*-induced liver tumors in tp53 mutations	*tp53* mutation reduces apoptosis and accelerates tumor progression; tumors regress after stopping the inducer	[20]

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
