# Peer review of "New Progress in Zebrafish Liver Tumor Models: Techniques and Applications in Hepatocellular Carcinoma Research"

_ijms, 2025, doi:10.3390/ijms26020780_

Round 1
Reviewer 1 Report
Comments and Suggestions for Authors
The manuscript aims to presents the review of novel developments in zebrafish models of hepatocellular carcinoma. The manuscript presents the data acquired from literature review with respect to different models mechanisms responsible for the tumor formation and techniques used to induce the model. The authors addressed both the mechanism of development and rationale of various models and their subsequent applications in scientific research. The authors critically described the cited articles in a comprehensive manner with strong emphasis on discussing strong and weak side of various approaches. Although some similar reviews were published concerning the topic (DOI: 10.3748/wjg.v21.i42.12042, https://doi.org/10.1016/j.jcmgh.2019.05.002), they were not as thorough and informational as the presented manuscript. The main strength of the manuscript are as follows: clearly presented and reviewed articles, structured and comprehensive approach to the review process, clear data presentation in form of tables and figures. The article is well written, the authors clearly draw a research gap and explained the shortcomings of various studies. I believe that it will be of interest for the readers of IJMS.
I would raise some points that could be better addressed by authors:
1) 36 out of 82 works are older than 5 years. In many cases this is understandable as the cited articles concern specific zebrafish developments, some more general citations surely could be replaced with newer sources, e.g. references 10, 13, 42.
2) In Section 2.4 the authors described the use of hepatocellular carcinoma models for drug screening. Indeed there are many problems associated with drug or potential drug candidates testing in zebrafish, e.g. different metabolism, drug delivery problems considering aquatic lifestyle of the zebrafish as well as biological barriers affecting drug absorption following immersion in drug solutions. Section 2.4 mostly focused on cachexia, but very little information was provided on actual drug screening. Could authors describe some valuable findings concerning the topic with emphasis on how the abovementioned challenges can be addressed in study protocol development?
3) In Section 2.6 it is unclear which environmental factors have previously been studied using zebrafish hepatocellular carcinoma models. Could authors provide some examples of such factors?
4) One of the challenges in zebrafish models is lack of well defined, detailed, structured and reliable procedures published in form of protocols. Could authors highlight examples of good practice in this regard in the Conclusion? This could provide a guidance for researchers reading this review.
5) In Section 4.1 sentence: ‘It was not until 2011 that stable transgenic zebrafish models capable of expressing liver cancer were reported.’ Should be followed by a reference to the report mentioned.
6) There are some unexplained abbreviations in Fig. 1 (MG, 4-HPPP), Table 3 (CACF, 4-OHT), Table 5 (4-ABP)
I would also highlight some minor points:
1) References 7 and 76 lack year of publication
2) Sentence on page 7: ‘In studies using the krasV12 transgenic zebrafish model, liver tumors were induced using tetracycline or analogs….’ should be changed to: ‘In studies using the krasV12 transgenic zebrafish model, liver tumors were induced using tetracycline or its analogs….’
3) In Section 4.4 a space is lacking in two sentences before reference [67]
Reviewer 2 Report
Comments and Suggestions for Authors
The manuscript provides a comprehensive review of zebrafish as a model organism for studying hepatocellular carcinoma (HCC). It discusses methods of model construction and their applications in studying tumor progression, metastasis, drug screening, and tumor regression. While highlighting zebrafish advantages like transparency and high-throughput screening, it acknowledges limitations, such as biological differences from humans. The authors conclude that zebrafish models complement traditional research tools and propose optimizing their use for enhanced clinical relevance in liver cancer studies. However, prior to be published, several considerations should be taken into account.
1. Enhance original contributions:
It will be great to include a section proposing new methodologies or innovative conceptual frameworks for integrating zebrafish models in liver cancer research.
2. Reduce Redundancy:
Avoid repeating points about the advantages of zebrafish (e.g., transparency and genetic similarity) across multiple sections. Instead, consolidate this information in a single, well-developed discussion.
3. About the Discussion on Limitations
The authors should provide a more critical analysis of the limitations associated with zebrafish models. Specifically, discuss the biological differences between zebrafish and humans, including metabolic and immune pathway differences, and how these might affect the translational relevance of the findings.
4. Discuss Translational Applications:
The manuscript should highlight how findings from zebrafish models can be applied to mammalian preclinical studies or clinical settings. This will bridge the gap between experimental research and real-world applications, providing a clearer roadmap for clinical translation.
5. Propose Future Directions:
Dedicate a final section to actionable future directions in zebrafish research. It could include combining zebrafish models with other animal models to overcome individual limitations or leveraging gene-editing technologies to create more sophisticated multitransgenic models.
Reviewer 3 Report
Comments and Suggestions for Authors
Dear Authors, an interesting review on zebrafish liver tumor models. I think minor revisions are required before the manuscript can proceed further. Please answer or consider the following:
(1) Abstract: try to change „this review” to „the present paper” at least once because the repetition is done three times. I suggest to change the middle one („Furthermore, this review explores …”).
(2) Introduction: I suggest putting Table 1 right after it was mentioned at the end of the second paragraph.
(3) Multiple locations: italicize „in vivo” and „in vitro” throughout the manuscript.
(4) All tables are hard to follow due to the word wrapping, line spacing and underutilization of the page area from left to right (tables are moved to the left). It is advised to improve their visualization because the content is appropriate.
(5) Section 2: there is no space mark at the end of the first paragraph (right before Figure 1).
(6) Figure 1: abbreviations must be explained, at least in the figure’s description. Especially those on the right side (MG, 419S1, etc). They are explained later in the text (in section 2.4), but their first appearance is here. I also advise you to mention in the description that the content of the graph will be thoroughly described in consecutive sections. Moreover, what did you want to visualize using the vials and circles on the left? If possible, please annotate it on the graph or explain it in the figure’s description.
(7) Section 2.2: change „destroy tumor cells” to „eliminate tumor cells”.
(8) When citing two or more papers I see that you use separate square brackets while I think it should be a single pair. For example, „[34, 35]” instead of „[34], [35]”.
(9) „HCC” is explained in the text multiple times. Please explain the abbreviation only on the first use of the full name.
(10) Section 2.6: if by „hMOF” you mean the „KAT8” gene, at least mention its current symbol. I also think you might switch the abbreviation and full name – move the latter to the brackets and the first outside brackets. Of course, this suggestion is optional. The fact of explaining this abbreviation only on first use and not using it afterwards seems to be unnecessary.
(11) Figure 2: are all groups of cells considered cancer cells? Because they are annotated above all graphs. Do you intend to show various cell groups? If yes, you can try to annotate them separately. If not, maybe put a single graph representing cancer cells?
(12) Figure 3: explain that „WT” stands for „wild-type” (e.g., in the figure’s description) or write its full name if it will not be used elsewhere.
(13) Section 4.2: in „low-KrasV12-expressing strains”, „KrasV12” is not italicized and is capitalized in comparison to other examples in the text.
(14) Figure 4: abbreviations such as „5-dpf” and „NASH” must be explained to Readers on first use (on the figure itself or in the figure’s description, unless it was mentioned above in the text). Currently, I think these abbreviations are after Figure 4 is shown.
(15) Section 5.3: „NAFLD” is explained but the first occurrence was in section 2.3.
(16) Figure 5: To improve Readers’ experience and attract a wider audience, explain abbreviations that were not mentioned earlier in the text, such as sgRNA, TSS, etc., even if their meaning is trivial to you.
(17) Conclusion: I would split the sentence when you provide three points after „the following”. Ideally, the part before points could be a separate sentence, and each point could also constitute its own sentence. This comment is optional.
(18) Abbreviations: delete the link next to „WHO” (if their inclusion was unintentional) and do not italicize „TERT” if other abbreviations are not italicized.
(19) Acknowledgments: there is probably a double space mark between „for” and „suggestions”.
